# Best of Both Hydrogel Worlds: Harnessing Bioactivity and Tunability by Incorporating Glycosaminoglycans in Collagen Hydrogels

**DOI:** 10.3390/bioengineering7040156

**Published:** 2020-12-02

**Authors:** Tanaya Walimbe, Alyssa Panitch

**Affiliations:** 1Department of Biomedical Engineering, University of California, Davis, CA 95616, USA; twalimbe@ucdavis.edu; 2Department of Surgery, University of California Davis Health, Sacramento, CA 95817, USA

**Keywords:** collagen, glycosaminoglycans, hydrogels, tissue engineering, hyaluronic acid, chondroitin sulfate, heparin, alginate

## Abstract

Collagen, the most abundant protein in mammals, has garnered the interest of scientists for over 50 years. Its ubiquitous presence in all body tissues combined with its excellent biocompatibility has led scientists to study its potential as a biomaterial for a wide variety of biomedical applications with a high degree of success and widespread clinical approval. More recently, in order to increase their tunability and applicability, collagen hydrogels have frequently been co-polymerized with other natural and synthetic polymers. Of special significance is the use of bioactive glycosaminoglycans—the carbohydrate-rich polymers of the ECM responsible for regulating tissue homeostasis and cell signaling. This review covers the recent advances in the development of collagen-based hydrogels and collagen-glycosaminoglycan blend hydrogels for biomedical research. We discuss the formulations and shortcomings of using collagen in isolation, and the advantages of incorporating glycosaminoglycans (GAGs) in the hydrogels. We further elaborate on modifications used on these biopolymers for tunability and discuss tissue specific applications. The information presented herein will demonstrate the versatility and highly translational value of using collagen blended with GAGs as hydrogels for biomedical engineering applications.

## 1. Introduction

Universal and ubiquitous in all multicellular organisms—collagen is essential to life as we know it. The triple helical protein not only amounts to the most abundant protein in the human body, but also plays countless roles in tissue structure, cell signaling, and modulation of cell behavior. There are over 29 forms of collagen found in mammalian tissues [1], of which type I, type II, and type III collagen are considered fibrillar collagens. As the main component of all connective tissues, fibrillar collagen accounts for about 55% of the dry mass of skin [2], 25–40% of the dry mass of articular cartilage, 70% of the dry mass of skeletal muscle [3], and 80–90% of the dry mass of bone [4]. It is also abundant in the cornea [5], blood vessels [6], the gut [7], and intervertebral discs [8]. Collagen type 1 is the most abundant fibrillar protein accounting for 90% of the collagen in the body and is composed of monomers containing three polypeptide chains that form a single right-handed triple helical structure [9,10,11]. It is these triple-helical monomers that assemble into collagen fibrils. In the body, collagen fibrils are crosslinked by the enzyme, lysyl oxidase, that acts on the ε-amine groups of lysine residues and supports crosslinks between modified lysine residues and also between modified and other lysine residues within collagen through aldol condensation reactions [12,13]. Crosslinking adds stability to the collagen fibrils and increases collagen tensile strength and resistance to enzymatic degradation [14]. The crosslinked fibers, composed of monomers assembled in a quarter stagger, show a characteristic visual banding pattern with a D-period of 67 nm [15]. Figure 1 shows the structure of collagen and its fibrils. The collagen structure aides in limiting tissue compliance, and also supports interactions with cells and other proteins and glycosaminoglycans. Thus it serves as both a key structural component and signaling component of the extracellular matrix.

The biophysical properties of collagen coupled with the relative abundance and ease of isolation from tissues including skin and tendon make it an interesting and popular biomaterial for tissue engineering, regenerative medicine, and drug delivery. The amino acid sequence of type I collagen consists of Gly-Xaa-Yaa repeats [16], with a substantial amount of Xaa and Yaa repeats consisting of proline (28%) and 4-hydroxyproline (38%) resulting from a post-translational modification of peptide-bound prolyl residues [17,18]. This atypical hydroxyproline content acts as a distinctive marker of collagen and is used in bioassays to identify collagen. In native collagen, the crosslinks generated through the action of lysyl oxidase are primarily found in the telopeptide regions of collagen, and these regions are mostly enzymatically removed from collagen during the process typically used to isolate collagen. As such, polymerized collagen in vitro is generally weak and low-density gels do not sustain their shape in the absence of an external crosslinker. Thus, for many applications researchers either crosslink collagen via one of a host of varying methods or add additional macromolecules to alter the biophysical properties of the hydrogels.

In addition to interacting with biomacromolecules, collagen acts as a ligand for cell receptors. The β_1_ integrin subfamily is responsible for facilitating binding to collagen. Specifically, α_1_β_1_, α_2_β_1_, α_10_β_1_, and α_11_β_1_ receptors bind to the GFOGER sequence in nondenatured collagen type 1 and binding is highly dependent on the presence of the glutamic acid within the ligand sequence [19]. Activation of integrins through collagen binding in turn causes a series of cellular outside-in downstream signaling events. Integrin activation leads to rapid activation of lipid kinases which promote the tyrosine phosphorylation of proteins such as focal adhesion kinase (FAK), p130Cas, and Src [20,21,22]. These then lead to the activation of signaling pathways such as Rho, Rac-1 GTPase as well as cytoskeletal proteins, which drive reorganization of the actin cytoskeleton [23]. These pathway activation responses manifest long term as changes in proliferation, differentiation, migration, and metabolism of cells [23,24]. The integrin-ligand interactions are important in tissue development, healing, and homeostasis as they represent a biophysical connection between the scaffold and the cell.

When collagen is denatured, such as in the form of gelatin, or in areas of damage including wounded or fibrotic tissue, collagen is partially denatured and can support α_v_β_3_ and α_5_β_3_ binding via normally inaccessible RGD sequences [25]. Given the importance of integrin activation in cellular processes, variation in activation of integrins can lead to unwanted consequences. In a careful study comparing cell binding and spreading in collagen, collagen and gelatin blends, and gelatin in monolayer form, in 2-dimensions on crosslinked gels, and in 3-dimensional culture, Davidenko, et al. demonstrated the importance of specific integrin binding in cell-collagen interactions. They also demonstrated the altering effect 1-ethyl-3-(3-dimethylaminopropyl) carbodiimide (EDC) crosslinking can have on these same interactions. As expected, they found that α_2_β_1_ and α_1_β_1_ integrin–collagen interactions dominated on fibrillar collagen while α_v_β_3_ and α_5_β_3_ dominated on gelatin. When cells are cultured in 3D, crosslinking with EDC reduces integrin binding to collagen and gelatin presumably though depletion of critical glutamic and aspartic acid residues within the binding regions. The decreased specific binding also came with increased non-specific interactions that may be due to changes in scaffold mechanics or chemistry [26]. Careful consideration thus needs to be given to integrin–ligand interactions for designing successful tissue-engineered scaffolds.

As researchers uncovered the innumerable roles played by collagen either as a building block or as a fundamental signaling protein, its popularity as a material for tissue engineering for restoring tissue function also surged. The versatility of fibrillar collagen—which could form the backbone of rigid tissues with a high Young’s modulus such as the bone, or provide support to compliant soft tissues with optical transparency such as the corneal stroma, or act as an elastomer with high shock absorbing properties in tissues such as the articular cartilage—truly made it a polymer with endless potential. Capturing these higher order supramolecular structures of collagen in vitro in hydrogels, however, has been challenging. Besides mechanical cues, cells are known to be sensitive to pore size and microstructure and alignment of the collagen fibrils [27,28,29,30]. Alongside, the environment in which collagen is polymerized such as ionic strength, temperature, pH, etc., imparts significant variability to hydrogel formation, structure, and properties [31]. The fact that type I collagen is a hallmark of scar tissue also warrants caution when using collagen to regenerate healthy tissue. This review aims to summarize the current strides made toward optimizing collagen hydrogels for biomimetic ECM-based applications.

We have included in this review only studies that construct hydrogels through polymerization of collagen monomers and multimers into the classical collagen fibrillar structure. Since gelatin is composed of denatured collagen and does not form fibrils despite forming hydrogels, to narrow the scope of the review to fibril forming collagen, we have not included studies with gelatin. Interested readers are directed to a review by Gorgieva and Kokol that discusses multiple modes of crosslinking and the effects on biophysical properties of both collagen and gelatin [1]. Further, we have not included studies that directly crosslink the acid solubilized collagen while still in its acid soluble form as this largely negates fibril formation. Similarly, if the form of the collagen hydrogel was not clear, the study was also not included. Finally, there are thousands of studies using collagen, and not all could be included in one review. We have attempted to highlight key aspect of collagen hydrogels, the benefits of including additional biopolymers within the collagen hydrogels, and the future perspectives.

## 2. Collagen Alone with No Crosslinkers

Collagen type I has interested engineers as the basis of hydrogels systems to culture cells in three dimensions for many years. This is in part due to the prevalence of collagen within the body, its ease of extraction, simple and cell-friendly assembly, and its biocompatibility. As previously noted, it also supports cell receptor interactions with ligands within the collagen triple helical structure making it an interesting material for cell culture for tissue engineering and for engineering tissue structures for in vitro drug testing.

Collagen hydrogels are synthesized by neutralizing acid-solubilized collagen using concentrated buffers [10× phosphate buffered saline (PBS), 10× cell culture medium, 10× Hank’s balanced salt solution (HBSS), etc.] to bring the ionic strength of the solution to 1× followed by the addition of neutralization agents (NaOH, HCl, and HEPES) and other reagents (water, 1× medium, 1× PBS) to initiate fibril self-assembly at a near physiological pH and polymerization temperatures of 37 °C. While this recipe for hydrogel preparation is universal, a closer look at the literature reveals that collagen hydrogel properties are highly dependent on collagen source (rat tail tendon, bovine skin, porcine, etc.) [32] and the method of extraction [33]. The hydroxylation of lysine and proline residues combined with lysyl oxidase crosslinking makes the production of functional collagen using traditional recombinant technology challenging. Researchers have therefore relied on extracting collagen from other species. Two main methods for extraction of collagen exist—acid solubilization at low pH values using an organic acid such as acetic acid, and a combination of salt precipitation with enzymatic extraction (pepsin digestion). Acid-solubilized collagen largely maintains its telopeptide regions that are critical sites for crosslinking, and in fact this extraction co-isolates a small number of multimers with crosslinks intact. On the other hand, pepsin digestion results in fully cleaved terminal non-helical regions that contain the intermolecular crosslinks. Pepsin-digested collagen results in a more soluble form of collagen resulting in higher yields, but also a form of collagen that polymerizes more slowly and exhibits decreased storage modulus values compared to acid-solubilized collagen [33,34] possibly since telopeptides play a strong role in fibril nucleation and are the locations of native crosslinks.

In a recent review investigating initial collagen solution concentration, pH, temperature, and ionic strength, it was noted that the rate of polymerization, compressive modulus, fibril diameter, and pore size in gels formed from acid solubilized collagen all influenced hydrogel properties [31]. Collagen concentration influences mechanical properties of the hydrogels, thereby influencing cell behavior [35,36]. This is perhaps not surprising when reflecting upon the fact that ECM compositions vary as does degree of crosslinking across tissue types. Predictably, reaction kinetics for collagen fibril assembly are temperature dependent, therefore, collagen molecules self-assemble more rapidly at higher temperatures. However, fibrils polymerized at higher temperatures show a lower number of bundled fibrils that are less ordered, consequently affecting the mechanical and structural properties of the hydrogel [37,38]. Fibril assembly initiates as soon as the collagen solution is neutralized regardless of temperature, hence, most groups work with collagen solutions on ice to slow the rate of polymerization until the hydrogel components are well mixed. pH of the solution also greatly affects structural and mechanical properties of the fibrils by modulating electrostatic interactions [39], thus adding complexity of working with collagen hydrogels. A strong positive correlation between pH and compressive strength exists; however, for physiological encapsulation, pH of hydrogels is limited to between 7.4 and 8.4 to maintain cell viability [40]. Both temperature and pH may influence the ratio of monomeric collagen to crosslinked multimeric collagen isolated from a different tissue type, or the same tissue type across species or from different aged animals or animals with a different environmental upbringing. Also, ECM composition variation may result in other types of collagen and accessory molecules co-purifying with the collagen type 1. All of these factors can influence the ultimate collagen hydrogel properties. Reviewers are directed toward the in-depth review by Antoine et al. for further reading on fabrication parameters [31]. A few examples for applications of collagen hydrogels synthesized through neutralization-based self-assembly follow.

High throughput drug testing is a necessity for the pharmaceutical industry and for drug delivery in general. However, a two-dimensional (2D) cell culture environment is highly artificial so does not provide for a highly reliable testing bed. For example, Millerot-Serrurot E et al. demonstrated that the 70% inhibitory effect of doxorubicin seen in 2D cultures was suppressed almost completely when cells were cultured in three dimensional (3D) culture in a collagen type I hydrogel [41]. 3D culture is one step closer to a realistic in vivo environment for drug testing, thus researchers have sought to create optically transparent, mechanically stable collagen platforms for this purpose [42]. Taking this a step further, Buchanan et al. developed a 3D in vitro microfluidic tumor vascular model under varying flow shear stress conditions [43]. Collagen hydrogels have therefore been established as a useful platform from which in vitro tissue for drug testing can be engineered.

While collagen type 1 is the most prevalent protein in the body, not all tissues contain large amount of collagen type 1. For example, articular cartilage and vitreous fluid contain predominantly collagen type II, but it is challenging to make structurally robust collagen type II hydrogels. Our lab therefore designed hydrogels by copolymerizing collagen type I and type II to harness the biological activity of collagen type II and the improved mechanical properties of collagen type I for articular tissue engineering [44,45]. Bioprinted collagen type II hydrogels with chondrocytes have also been designed to form biomimetic hydrogels with cell density gradients to mimic cartilage zones [46]. However, ligaments, tendons, skin, and bones contain a preponderance of collagen type 1 as compared to other proteins in the tissue. As such collagen type I hydrogels have been studied as the base materials for vast majority of collagen hydrogel-based tissue-engineered and tissue-repair studies, especially for tissues rich in collagen. Some examples of the use of collagen hydrogels in tissue engineering follow, although these examples are demonstrative rather than exhaustive.

Collagen has served as a popular scaffold for culture and differentiation of mesenchymal stem cells (MSCs). Zhang et al. demonstrated that collagen hydrogels can be used in the absence of external stimuli such as transforming growth factor-β (TGFβ) to encapsulate MSCs and induce chondrogenesis [47]. While these results are impressive, it is commonly appreciated that just collagen alone is insufficient to maintain long-term differentiation. It is likely that additional additives will be necessary to sustain differentiation and obtain fully functional cartilage. Similar findings are prevalent with MSCs and osteoblast differentiation, where studies suggest that collagen gels support early differentiation [48]; however, collagen gels alone have not been shown to support long-term differentiation, especially following implantation. To build on the usefulness of collagen as a base structure from which conditions that support both early and stable MSC differentiation can be developed, collagen hydrogels were used to investigate the effects on fibroblast growth factor (FGF) on MSC differentiation into osteoblasts. Ultimately, 5 ng/mL of FGF was found to be the minimal FGF concentration necessary to maximize cell proliferation and alkaline phosphatase activity, providing strong evidence that additional bioactive factors can help to support early and ultimately stable differentiation [49]. The apparent requirements for additional bioactive agents to support stable differentiation is not specific to collagen hydrogels. Similar limitations are seen with biomaterials generally, and as with collagen, are likely due to the minimal signals provided by the scaffold. However, added stimuli do not need to be chemical in nature. Using mechanical stretching during MSC culture in collagen hydrogels, Noth et al. developed tissue structures with fairly oriented collagen fibrils reminiscent of those seen in native tendons and ligaments. In the absence of stretch, the collagen hydrogels did not support this same aligned tissue structure, but formed a tissue with a more random fibrillar morphology stretch [50]. Finally, collagen hydrogels have also been used to support the development of cancer tumors in vitro [51]. Tumors are formed that have hypoxic cores and that produce expected molecules such as hypoxia inducible factor 1 and vascular endothelial growth factor 1. The more successful use for in vitro tumor development is perhaps not surprising since in many tumors, the collagen is not well structured and is highly prevalent just as it is in many hydrogels.

Additional limitations exist. Collagen hydrogels, when polymerized in the absence of applied external forces, form a swollen network of randomly oriented collagen fibrils. However, in the ECM of many tissues, collagen fibrils appear highly oriented, and this orientation can vary with tissue type and location within a tissue. For example, in articular cartilage, the superficial zone, the zone closest to the articular surface, has fibrils oriented paralleled to the tissue surface, while the zone closest to the bone contains fibrils oriented perpendicular to the surface [52]. Similarly, corneal stroma, which consists of type I and type V collagen, exists in parallel organized fibrils that are further arranged into orthogonal lamellae, and considered necessary for the optical transparency of the cornea [53]. Thus, it appears that fibril orientation may be tissue instructive with respect to biophysical properties including strength, and cellular communication. Several studies have investigated ways to orient collagen including applications of electric field to take advantage of the collagen isoelectric point focusing [54,55], applications of magnetic fields [56,57,58], and flow [59]. The important role of mechanics in fibril orientation was recently reviewed [60]. All of these processes support collagen fibril assembly complete with expected D-banding. To highlight the importance of fibril orientation, strain-induced alignment of collagen fibrils improved nerve regeneration. As seen with other polymers, alignment helped to guide neuronal outgrowth in the direction of fibril alignment and promoted longer extensions as compared to those seen in unaligned collagen [61]. While not performed on hydrogels, but on densely packed electrochemically induced aligned collagen fibrils, studies showed improved MSC differentiation into tenocytes as compared to the differentiation seen on randomly aligned fibrils, again showing the importance of orientation [55]. Accessory molecules, like the small leucine-rich proteoglycan decorin, have been extensively documented for their ability to control collagen fibril alignment in vivo emphasizing the importance of the biophysical aspect of collagen as researchers seek to further improve the outcomes when using collagen as a scaffold [62].

While collagen gels possess many positive attributes including the ability to conform to defects, to gel in the presence of cells, and to support remodeling by cells, a significant shortcoming is the mechanical weakness of the gels. Soft collagen hydrogels do not support significant weight bearing and are challenging to handle because of shape change and tearing when moved. For this reason, researchers have copolymerized collagen with other macromolecules, crosslinked collagen by various means, formed high concertation gels and even compressed collagen hydrogels to form dense hydrated mats. A study by Helary et al. suggests that at concentrations above 3.0 mg/mL collagen gels become easier to handle and could be transferred via spatula from one dish to another; although the storage modulus (290 ± 54) indicates that the gels remain relatively weak [63]. However, as the collagen concentration increased, fibroblasts compacted the gels to a smaller extent demonstrating that polymer concentration in the absence of crosslinking affects the cell behavior. Whether this is through additional points of interaction with the cells, stiffer matrices with which the cells are interacting, or a combination of the two remains to be elucidated. Collagen self-assembly and the ability to encapsulate cells was capitalized upon to form thin mats for skin tissue engineering. Compression of the cell-laden collagen gels to approximately one tenth their starting volume led to development of hydrated sheets that could withstand mechanical manipulation [64]. The fibroblast-embedded collagen sheets proved to be a viable strategy for engineering skin grafts. Similar efforts using compression to increase fibril content and thereby introduce a controlled spatial gradient of oligomeric collagen for revascularization of skin wounds have been undertaken [65]. As with these studies, going forward it will be key to balance cell-material interactions and desired biological outcome with needed handling properties of the material to construct a functional living hydrogel.

Despite success seen with compacted hydrogels, significant physical shortcomings remain with the use of type I collagen alone as a scaffold; therefore, researchers have attempted to crosslink collagen or polymerize it with other relevant macromolecules to improve its mechanical properties and applicability as a construct to engineer specific tissue types. The following sections elaborate on efforts to use crosslinked collagen as well as blend collagen with glycosaminoglycans (GAGs), which are carbohydrate biopolymers found in the ECM.

## 3. Collagen with Crosslinkers

On the one hand, crosslinking collagen can improve its tunability, strengthen its mechanical properties, and reduce its degradation. However, having mentioned the importance of ligand-receptor interactions for cell signaling and cohesive tissue responses, careful consideration needs to be given to preserving native adhesion sites and fibril formation and not disrupting biocompatibility through crosslinking. Hence, choosing the lowest degree of modification that provides the necessary mechanical properties is usually recommended. The majority of chemical crosslinking in collagen is achieved by targeting either amine groups or carboxylic acid groups in amino acid side chains—(1) through covalent amine linkage by targeting the ε-amino group of lysine and hydroxylysine, or (2) through carboxylic acid groups found in aspartic acid and glutamic acid residues. Reviewers are directed to a recent review that describes the different natural and chemical crosslinking agents for collagen and their chemistries in depth [66]. Of the studies that have tested different crosslinkers, primarily found to be nontoxic, in order to tune the mechanical and biological properties of collagen and stabilize it for downstream applications, EDC/N-hydroxysuccinimide (EDC/NHS), glutaraldehyde, and genipin are most common (Figure 2).

Crosslinked collagen hydrogels are a popular material for corneal tissue engineering. Goodarzi et al. crosslinked collagen–gelatin blends or gelatin alone using EDC/NHS chemistry. They found that collagen improved transparency, mechanical stiffness, swelling, and pore size while decreasing the susceptibility to enzymatic degradation. In addition, MSC proliferation and integration into the gel over time was improved in the presence of collagen [67]. Han et al. constructed a layered collagen membrane using EDC-crosslinked collagen type I and type II for cartilage tissue engineering and showed good biocompatibility of the gels [68]. Despite promising results, care must be taken with EDC since it is added to activate crosslinking, but not consumed to a great amount during the reaction; residual EDC can have unintended biological consequences in addition to altering the primary structure of collagen. Furthermore, if too much EDC is added, it can consume excess carboxylic acid sites through O-acyl to N-acyl urea conversion and can alter the bioactivity of the collagen hydrogel. In fact, Davidenko et al. studied this very phenomenon and found that in many studies more than 100× EDC was used as compared to that needed to achieve desired mechanical properties [14]. Another major drawback of EDC is the fact that while EDC and NHS facilitate crosslinking of carboxylic acids and amines in collagen, they need to be washed away from the scaffold, thus limiting their applicability for in situ crosslinking with cells and in vivo use.

As mentioned with respect to EDC, crosslinking can improve collagen hydrogel mechanical properties and stability, but it can also alter other properties due to consumption of functional groups on the collagen backbone, or creation of new functionality. In the case of genepin, crosslinked collagen gels can become optically purple or black depending on the degree of crosslinking. Genepin crosslinking also alters the inherent fluorescence of collagen and the structure of the collagen fibrils formed as observed through altered striation patterns within the fibrils [69]. Despite this shortcoming, genepin remains a popular choice for crosslinking collagen. To study the effects of different cross-linking conditions of genipin on type I collagen scaffolds, Zhang et al. polymerized type I collagen in the presence of different concentrations of genipin, as well as different polymerization temperatures [70]. Based on their studies, a concentration of up to 0.3% genipin and a temperature of 37 °C was found to be optimal for crosslinking. Along similar lines, another study reported that low concentrations of genipin can be safely used for crosslinking collagen to improve its mechanical properties without affecting cell viability [71]. Whether these results translate well in vivo remains to be seen. Crosslinking can also be useful when trying to use bioactive polymers that do not form stable hydrogels by themselves. For example, genepin has been used to crosslink type II collagen, which does not inherently polymerize well in vitro [72]. In this way, weaker fibrillar collagen types such as type II and type III can be incorporated in scaffolds to harness their bioactivity.

Glutaraldehyde has also been used to crosslink with amine groups on lysine and hydroxylysine residues in collagen [73,74,75]. However, cytotoxicity from residual glutaraldehyde and over-crosslinking limit its application in modern day hydrogel scaffolds [76,77].

Thiolation of gelatin is a popular method to add thiols to gelatin for future crosslinking with other thiols to form disulfides or with acrylates through Michael type addition [78]. Along those lines, researchers have attempted to thiolate collagen to use the functional thiol for controlling the degree of crosslinking and increasing the stability of collagen hydrogels by adding other biopolymers like polyethylene glycol (PEG) or bioactive compounds to it. There are multiple ways to introduce thiols and disulfide linkages in collagen. One method to form thiolated collagen is by reacting collagen with succinic anhydride to yield carboxylated collagen (Col-COOH), followed by amidation with 2-mercaptoethylamine hydrochloride (MEA) [79] or to simply react with N-succinimidyl S-acetylthioacetate [80]. Another method is reacting it in an imidazole aqueous solution or dimethyl sulfoxide and reacting with γ-thiobutyrolactone [81,82]. Thiolating collagen provides the opportunity to tune collagen hydrogels to the required tissue while improving its stability. However, it is unclear the degree to which the collagen retains its fibril-forming abilities upon modifications with thiol-containing agents, as microstructural characterization was not performed in these studies. The Tanabe research group showed that disulfide crosslinked collagen could preserve partial helix structure [83], suggesting that this modification reduces the fibril-forming ability of collagen. Future studies characterizing the effect of loss of fibril formation on biocompatibility will be needed to provide insights on the degree of modification ideal for use in hydrogel systems.

Apart from chemical crosslinking, researchers have also used physical crosslinking methods such as dehydrothermal (DHT) treatment [84,85] and UV irradiation to crosslink collagen [86,87,88,89]. DHT treatment results in amide bond formation between adjacent carboxylic and amine groups through a condensation reaction. A study by Cornwell et al. compared the effects of DHT vs. chemical crosslinking on hydrogel mechanical properties as well as cell migration [90]. While results showed that all crosslinking techniques increased the tensile strength of the hydrogels and decreased enzymatic degradation rates in comparison to non-crosslinked scaffolds, DHT crosslinks most significantly improve mechanical strength, but also reduce cell migration the most. This suggests that chemical crosslinking using EDC is milder and might offer greater control over properties in comparison to DHT or UV irradiation-based crosslinking. UV irradiation also leads to concerns about partial denaturation of collagen and raises questions about reduced biocompatibility.

While great strides have been made in understanding collagen as a material for hydrogel formation, collagen itself has several limitations. As noted, collagen polymerization is highly dependent on environmental conditions, the resultant gels have unoriented fibrils unless external stimuli are applied, and the gels are mechanically weak unless polymerized at high density or crosslinked with exogenous crosslinkers, which can impair the biological activity. Additionally, collagen type 1 is often found densely packed in scar tissue, and semi-denatured or damaged collagen type 1 is found in damaged tissue and can contribute to scarring. Finally, in the body, collagen is found within a milieu of additional extracellular matrix molecules and signals. Thus, the evolving chapter of developing an understanding of collagen hydrogels and their use in tissue healing and regeneration comes in the form of blended gels. Of focus for the remainder of this review is specifically the incorporation of GAGs into collagen hydrogels as they represent a critical component of collagen-rich tissues.

The ECM of most tissues is heterogenous and dynamic and balances hydrophilic and hydrophobic moieties to maintain homeostasis. The ECM consists of two main classes of macromolecules-fibrous and amorphous proteins and polysaccharides in the form of glycosaminoglycans (GAGs) commonly attached to a core protein to form proteoglycans. Scientists have moved past using collagen alone in hydrogels to create more physiologically relevant hydrogels for 3D culture as well as treatments. Given that GAGs are also ubiquitous and almost always seen in conjugation with collagen in the ECM, researchers have logically diverted to improve collagen hydrogel tissue engineering by incorporating GAGs into them in order to generate blended ECM mimetic and tunable materials. While GAGs are more tunable, water soluble, and hydrophilic than collagen, with the exception of hyaluronic acid, they do not provide known high affinity sites for adhesion of cells. Blending GAGs with collagen therefore mutually augments the copolymerized hydrogel properties to overcome their individual disadvantages and provides the opportunity to engineer more physiologically relevant, synergistic hydrogels.

## 4. Collagen–GAG Hydrogels

Early studies with collagen hydrogels including GAGs set the foundation for the importance of GAG-collagen blends in affecting biophysical properties of gels and influencing cell behavior. For example, in 1989, Docherty, et al. reported that inclusion of GAGs in collagen gels supported enhanced fibroblast motility [91]. Further, they shed light on the importance of GAG identity by demonstrating heparin-inhibited collagen polymerization, while low concentrations of HA and CS enhanced cell migration rates, with HA doing so more effectively, while low MW HA was more effective than higher MW HA [91]. In 1996 Bitner et al. demonstrated that decorin altered the ability of fibroblasts to contract collagen gels [92]. These and other studies peaked researchers interest in combined collagen-GAG formulations.

There are six different types of GAGs which are classified based on their disaccharide monomer repeats—hyaluronic acid (HA), heparin, heparan sulfate (HS), chondroitin sulfate (CS), dermatan sulfate (DS), and keratan sulfate (KS) [93]. GAGs are polymer chains composed from disaccharide monomer repeats of glucuronic acid (GlcA) or its epimer iduronic acid (IdoA), and N-acetylgalactosamine (GalNAc) or N-acetylglucosamine (GlcNAc) (Figure 3) [94]. They are post-translationally modified to have variable sulfation patterns that lead to their added complexity and highly negative charge [95]. As with collagen, GAGs are not merely a structural component of the ECM, they also perform a diverse range of functions to modulate tissue homeostasis. GAGs are known for the anti-inflammatory and protective role they play because of their ability to mask cells and other proteins to prevent activation of immune cells—for example, GAGs are a major component of the endothelial glycocalyx, which when stripped, leads to the exposure of ligands that bind and activate neutrophils, monocytes, and platelets [96,97,98]. Additionally, since GAGs are negatively charged, they interact with and attract water to load bearing tissues such as the cartilage and vocal folds, thus increasing their compressive strength and providing shock-absorbing capacity to these tissues [99,100]. This makes them all the more attractive as components of blended hydrogels. Alongside, since they do not contain a protein component, they exhibit minimal antigenicity and demonstrate excellent biocompatibility for tissue engineering purposes. Moreover, modifications can easily be introduced into GAGs through the carboxylic acid group present on the backbone, and more importantly simple modifications can be used to tailor GAGs for specific applications based on their functionality.

As with crosslinked collagen hydrogels, parameters such as concentration, crosslinking density, and retention of bioactivity are important fabrication parameters for the design of collagen-GAG hydrogels. High concentrations of GAGs inhibit fibril formation [101], limiting the amount of GAGs that can be added to the hydrogel without negatively affecting the fibril formation. Degree of modification and crosslinking density directly correlate with increased stiffness and stability of the hydrogels, which can be used to design hydrogels to match tissue microenvironments. However, care needs to be taken to not over-substitute GAGs, since higher degrees of modifications result in the loss of bioactivity [102]. Therefore, a balance between concentration, modification degree, and bioactivity needs to be found for the design of GAG-collagen hydrogels. Designing successful hydrogel candidates consequently requires careful understanding of the required tissue outcome and parameters important for regeneration. The next section includes examples of collagen-GAG hydrogels organized by the type of GAG used.

### 4.1. Collagen–HA Hydrogels

HA is the only non-sulfated GAG and is the most widely used GAG for tissue engineering because of its ease of handling and modification, excellent biocompatibility, diverse molecular weights that can be used to tune its biological and mechanical properties, and high swelling capacity [103]. Hydrogels with HA and collagen do not bind and modulate growth factors to the same extent as those containing sulfated GAG, and in turn, do not affect cell behavior to the same degree. In order to increase the ability of HA-collagen hydrogels to bind growth factors to sustain regeneration, researchers have chemically modified HA to contain sulfate groups and increase its negative charge [104,105,106]. Hydrogels containing sulfated HA could have especially promising applicability for treating chronic wounds and inflammation due to their ability to sequester and release growth factors while modulating cell behavior. Another area of interest is to use these growth factor laden hydrogels to guide stem cell behavior and drive differentiation based on the presentation of growth factors in the hydrogels [107,108,109,110]. Studies have used this approach to guide osteogenic differentiation of MSCs without the use of steroids like dexamethasone [108]. Collagen hydrogels contain modified and unmodified HA have shown great promise as tissue engineering and drug delivery scaffolds.

Thiolated HA is a popular material for hydrogel formation since the thiol groups can be used for both crosslinking and for conjugation of additional functional molecules. Thiolated HA has been extensively used with crosslinked gelatin and poly (ethylene glycol) diacrylate (PEGDA) for close to two decades [78,111,112]. Other researchers have taken the route of modifying the collagen with methacrylate groups to crosslink with the thiolated HA [113]. However, to avoid the drawbacks inherent to non-fibrillar gelatin and crosslinked collagen, namely the exposure of normally cryptic integrin-binding sequences, the loss of integrin-binding sequences exposed on fibrillar collagen, as well as potential loss of key integrin ligands when crosslinking collagen, our lab designed interpenetrating (IPN) hydrogels with fibrillar collagen type I and type III entrapped within crosslinked thiolated HA. HA crosslinking was achieved using PEGDA, and development of an IPN was achieved by modulating the pH of the hydrogels during polymerization to drive collagen polymerization faster than the Michael type addition between thiols and acrylates [114]. Interpenetrating collagen fibrils could be visualized in the resulting hydrogel, and the gels showed good biocompatibility in culture over 21 days. In this way, tunable hydrogels can be formed without altering the structure of the collagen.

IPNs can also be formed using free radical polymerization. Ultraviolet light (UV)-induced free radical generation offers the advantage of crosslinking acrylated or methacrylated HA after collagen fibrillogenesis is complete to form IPNs without the need to modify the collagen [115,116,117]. Hydrogels synthesized using glycidyl methacrylate HA or methacrylic anhydride modified HA and collagen formed IPNs that slowed degradation of the scaffolds due to UV crosslinks, improved mechanical properties of the scaffolds, and allowed for encapsulation of cells without being cytotoxic. These hydrogels are easy to pattern since the precursor can be patterned and polymerization commenced with exposure to UV light. These gels also provided attachment to various cell types such as Schwann cells and fibroblasts, demonstrating their versatility for biomedical applications. However, the use of a photointiator for UV crosslinking has limited clinical application because of the risk of free radical-induced cell damage and toxicity from the initiators used.

EDC crosslinked HA-collagen hydrogels have been explored for many applications including dermal, corneal, neural, and cartilage tissue engineering, as well as wound healing, etc. Unlike the Collagen HA IPN described above, by using EDC the collagen is crosslinked directly to the carboxylate groups on HA. One advantage of forming collagen-GAG blend hydrogels is the ease with which crosslinker concentration can be used to control hydrogel stiffness while maintaining collagen and GAG biological activity. In an interesting study evaluating the effect of stiffness on stem cell lineage, Her et al. designed HA-collagen scaffolds with stiffness varying from 1 kPa to 10 kPa by varying the amount of EDC-supported crosslinking, and found that they could drive hMSC differentiation toward a neuronal lineage on softer substrates and a glial lineage on the stiff substrates [118]. Murphy et al. found a similar effect while attempting to differentiate MSCs into chondrogenic vs. osteogenic phenotypes, with the amount and crosslinking of GAGs to the collagen influencing outcomes [119]. Other studies have also shown that the addition of HA to collagen scaffolds improves chondrogenic differentiation of MSCs [120,121,122], which is not surprising, given that the articular cartilage consists of ECM rich in GAGs and collagen for shock absorption and mechanical loading properties.

Because all GAGs contain carboxylate groups that can be activated by EDC and other agents for crosslinking to free amines on collagen fibrils, it is possible to form collagen hydrogels with blends of GAGs. For example, in studies aimed at designing scaffolds for dermal repair, Wang et al. crosslinked HA and CS with collagen using EDC to form nine different blends with varying ratios of collagen:HA:CS [123]. Scaffolds seeded with allogenic fibroblasts in a 9:1:1 blend of crosslinked Col:HA:CS showed almost normal skin 6 weeks after implantation in Sprague-Drawley rats in comparison to controls, indicating their potential for regenerating skin tissue. GAGs can also be used to control the optical properties of collagen gels, which when polymerized alone are not optically transparent. In another study, EDC/NHS crosslinking resulted in collagen-gelatin-HA films with optical performance, hydrophilicity, and mechanical properties suitable for corneal tissue engineering [124]. Further, these same gels were used to design a wound-healing scaffold for chronic wounds, since EDC crosslinking increases biological stability by decreasing enzymatic degradation in the highly proteolytic chronic wound environment [125]. Collectively, these studies demonstrate the importance of modulating the degree of crosslinking and biological activity to achieve intended biological outcomes.

One limitation to the EDC crosslinked materials described above is the development of largely elastic hydrogels. To circumvent problems associated with these purely elastic materials, the Chaudhuri group designed IPNs of HA and collagen by mixing HA-hydrazine, collagen, and HA-aldehyde (or benzaldehyde) solutions to form stress relaxation hydrogels that contain dynamic covalent bonds [126]. They showed that stress relaxation promoted cell spreading, fiber remodeling, and focal adhesion formation in 3D, and could guide mechanotransduction and recapitulate the fibrillar collagen architecture of many cellular microenvironments.

While EDC is a popular choice to induce crosslinking, other agents have also been studied to crosslink GAG-collagen gels. GAGs, including HA can be readily modified to contain aldehyde groups, however reactions between aldehydes and amines results in unstable crosslinks. Oxime reactions have the advantage of being more stable than Schiff bases formed between aldehydes or ketones and amines, while providing a similar level of tunability. Hardy and Schmidt used oxime-bond click chemistry to couple aldehyde group containing HA with aminooxy-terminated poly (ethylene glycol) (PEG), and further incorporated collagen type I in these gels to provide adhesion sites for MSCs to aid neural tissue regeneration [127]. The chemistry was not toxic to primary Schwann cells, and the hydrogels could be tuned to have mechanical properties analogous to those found in soft tissues such as the central and peripheral nervous system, demonstrating the wide breadth of tissues HA-collagen hydrogels can be tuned towards. Some studies have also attempted to use adipic acid dihydrazide (AAD)-modified HA to crosslink with collagen hydrogels. Genepin crosslinking of AAD modified HA and collagen have shown increased stability and retention of HA in these composite scaffolds [128], possibly because of slowing of degradation due to the crosslinked nature of the hydrogels.

Overall, this vast array of studies shows that the addition of crosslinked HA improved handling and biological outcome of the hydrogels over those seen with collagen alone. Collagen-HA hydrogels have wide-ranging applicability for tissue engineering due to their superior ability to be modulated based on the target tissue biophysical properties. As with any crosslinked gel, the inverse correlation between bioactivity and tunability will dictate the efficacy of the scaffolds for tissue engineering applications. The added complication of balancing hydrophilicity and hydrophobicity to ensure polymerization of collagen in the water-loving HA provides some design challenges that need careful consideration, since a high concentration of GAGs, including HA, will inhibit polymerization of collagen. Therefore, along with degree of modification, concentrations of the collagen and GAGs along with the pH of the solution will be crucial in designing successful constructs with the desired biophysical properties. The next generation of treatments will potentially involve the use of these engineered matrices in conjugation with growth factors and cells for sustained regeneration of tissues.

### 4.2. Collagen–CS Hydrogels

As the most abundant GAG in the body, CS is found in the vitreous fluid, the glycocalyx and other connective tissues, but is found in highest concentrations in the cartilage. It is a sulfated GAG that is known for its shock absorbing properties, maintenance of cartilage structure and function, and prevention of inflammation, due to which it has been extensively used for the treatment of osteoarthritis [129,130,131].

Like with HA, the use of EDC/NHS chemistry has been routinely performed to crosslink CS to collagen for hydrogel formation due to its low cytotoxicity and ease of handling. However, even with its extensive use, crosslinking Collagen-GAG hydrogels with EDC is not trivial, especially when the goal is to form interpenetrating networks. The van Kuppevelt group conducted extensive studies to optimize crosslinked CS-collagen based hydrogels. In an initial study, two different types of crosslinking techniques, namely, DHT and EDC crosslinking were investigated as a means to incorporate CS in type I collagen hydrogels [132]. EDC crosslinking of CS with collagen pre-crosslinked using DHT treatment resulted in collapsed matrices with CS decorating only the exterior of the collagen, possibly due to the already crosslinked and most probably denatured nature of the collagen. EDC crosslinking to non-crosslinked collagen under aqueous conditions also led to partial collapse of the matrix, which could be circumvented in the presence of ethanol. The addition of an organic solvent such as ethanol likely suppresses urea forming side reactions that commonly occur in water and promote amide bond formation between collagen and GAGs, resulting in evenly distributed crosslinked hydrogels. However, the addition of ethanol precludes the polymerization of these scaffolds in the presence of cells in vitro and limits that potential to implant cell-embedded scaffolds in vivo. They were however able to test the bovine type I collagen with CS or HS crosslinked using EDC-NHS in the presence of ethanol in rats (post ethanol removal) to evaluate tissue response to these hydrogels. The addition of GAGs reduced foreign body reactions, which in turn reduced degradation of the scaffolds in comparison to purely collagen implants. The highly negative charged scaffolds containing GAGs also led to enhanced angiogenesis, possibly through sequestration of growth factors such as FGF, vascular endothelial growth factor (VEGF), platelet derived growth factor (PDGF), etc. [133].

When the goal is not to form an IPN, but a CS-collagen gel, EDC crosslinking can be beneficial. For example, Gao et al., used CS activated with EDC NHS (CS-sNHS) to form crosslinked hydrogels with type II collagen to use as a cell delivery system to treat defects in the articular cartilage [60]. Crosslinking type II collagen with the CS is an innovative way to get type II collagen to stay in the hydrogel, since it is not capable of forming strong fibrils spontaneously in vitro. The scaffolds showed good biocompatibility and were tunable depending on the amount of CS-sNHS added. Several additional studies have demonstrated improved outcomes with the use of collagen-CS scaffolds for cartilage tissue engineering. While these studies approached the incorporation of CS in collagen hydrogels using different crosslinking methods such as via methacrylated CS or using genipin, the end goal was similar-to promote chondrogenesis and tune the mechanical properties of these hydrogels to suit articular cartilage tissue engineering [129,130,131]. The next generation of cartilage therapy will likely include hydrogels with encapsulated cells for superior and sustained regeneration of the cartilage.

EDC/NHS crosslinked CS-Collagen 3D sponges have also been used for applications including neural tissue engineering applications and embryonic neural cell culture, and showed superior thermal resistance and lower enzyme sensitivity than collagen alone [134,135]. This shows the flexibility and promise of CS-Collagen systems for tissue regeneration.

It is also possible to incorporate GAGs into collagen in the absence of additional chemical crosslinking. Stuart and Panitch demonstrated that the addition of CS to collagen type I led to network organization with an increase in void space and decreased stiffness of the gels [136]. Since type II and type III collagen are also a major ECM component in some tissues such as the articular cartilage and vocal folds, researchers have sought to incorporate these fibrillar collagens in an attempt to make more biologically relevant hydrogels. Studies in our lab have shown that polymerizing type I and type III collagen together with CS leads to a more open and compliant hydrogel network [137].

CS–collagen hydrogels thus provide an exciting class of hydrogels which can be used to reduce collagen compaction, modulate cell response, and drive differentiation of stem cells while allowing the same flexible level of tunability offered by collagen–HA hydrogels. Taken together, the benefits of adding CS to collagen with respect to amplifying the bioactivity and cell instructive nature of the hydrogels further suggests that the addition of GAGs to collagen can improve tissue outcomes.

### 4.3. Collagen-Heparin Gels

Unfractionated heparin is best known for its anticoagulant properties and commonly used as a blood thinner [138]. It is the most negatively charged GAG, and thus demonstrates binding and sequestering of growth factors along with being anti-inflammatory. This ability can be exploited to design bioinstructive hydrogels that are capable of providing sustained release of growth factors along with the benefits of the biocompatible materials themselves. Copes et al. exploited this ability of heparin to engineer collagen–heparin hydrogels for the controlled release of pleitropin for vascular applications [139]. Though studies have shown that high concentrations of heparin can inhibit collagen fibrillogenensis [140,141], no effect of heparin on collagen fibril formation was seen in this study. This strategy of providing cell instructive cues and entrapping growth factors can be useful for sustained release of many growth factors to provide persistent regeneration [142]. Another attractive area of application is to guide controlled differentiation of stem cells by sequestering relevant growth factors for augmenting tissues [143].

In other uses, maleimide functionalized heparin linking to star-PEG, has been used with collagen to form hydrogels containing cell instructive peptides as an innovative way to combine semi-synthetic and native ECM molecules to closely mimic the ECM [144]. Modifying the heparin to add functionalities can similarly be applied in the future to include cues for regeneration and repair. While unfractionated heparin is anticoagulating, low molecular weight heparins can be substituted to pivot to applications where anticoagulation is not a desirable outcome. Combined with the growth factor sequestering ability of heparin, this creates additional exciting venues for regenerative medicine applications.

### 4.4. Collagen-Alginate Gels

Alginate is a naturally occurring anionic carbohydrate polymer obtained from seaweed. While not bioactive like other GAGs and not a traditional component of the ECM, alginate has been widely used for tissue engineering because of its biocompatibility, low toxicity, relatively low cost, and tunability. Sodium alginate forms a hydrogel in aqueous solutions by divalent cation binding [145]. This makes it a highly versatile and tunable GAG analog that can be used as a building block in hydrogels. Another major advantage is that the degree of alginate crosslinking will not affect collagen adhesion sites, providing a way to engineer tunable scaffolds without significantly affecting the bioactivity of collagen fibrils.

To model in vitro tumor environments that offer flexible control of mechanical and biophysical features, Liu et al. engineered composite alginate-collagen hydrogels. Hybrid alginate-collagen hydrogels can be tuned to demonstrate a wide range of elastic moduli by changing the concentration of CaCl_2_ crosslinker, and used to model tumor invasion of breast cancer cells [146]. Moxon et al. blended collagen with alginate to mimic the hyaluronic acid and collagen-rich environment of the brain ECM [147]. The substrate was conducive to the growth of human iPSC-derived neurons, and tunable based on the amount of ionic crosslinker added for gelation of alginate. Comparing different GAG blends with collagen for vocal fold tissue engineering, collagen-alginate composite hydrogels were shown to resist scaffold compaction and mass loss for at least 42 days in culture while allowing for ECM synthesis in comparison to collagen-HA composite hydrogels, thus showing that the type of GAG used can influence not only gel properties, but also cellular outcome [148].

Several studies have used collagen-alginate hydrogels as biocompatible substrates for chondrocytes geared toward tissue engineering of the articular cartilage [149,150,151]. Yang et al. employed 3D bioprinting to precisely print collagen-alginate gels for articular cartilage, since bioprinting can provide improved control over spatial and architectural orientation [151]. The high water retention capacity and reduction of gel contraction in composite collagen-alginate hydrogels makes it an attractive hydrogel for articular cartilage tissue engineering in comparison to collagen alone.

In the space of drug delivery, alginate hydrogels have been used as carriers for combinatorial photothermal and immuno tumor therapy by simultaneous encapsulation of the photothermal drug methylene blue (MB) and immunological agent imiquimod (R837) for prolonged and sustained delivery [152]. In a separate study, Lee et al., encapsulated glial cell line-derived neurotrophic factor (GDNF) secreting HEK293 cells in collagen-alginate microspheres for the controlled release of GDNF. They further went on to test these cell encapsulated microspheres for the treatment of neurodegenerative posterior eye diseases in a rat model and showed promising results with improved photoreceptor survival in dystrophic rat eyes [153].

While not a native GAG, alginate continues to grow in popularity because of its mild gelation conditions, biocompatibility, and tunability. Unfortunately, release of the divalent ions responsible for crosslinking the gel causes alginate gels to dissolve, limiting the long-term stability of alginate-collagen gels in physiological conditions. This can be beneficial or negative depending on the application, but needs to taken into consideration before designing the hydrogels. Alongside, alginate plays a passive role in the hydrogels by not modulating signaling directly. Collagen-alginate hydrogels thus make for attractive carriers for drug delivery. Future studies will likely look at incorporating bioactive factors and targeting peptides to increase alginate functionality for drug delivery applications.

## 5. Conclusions

As seen from the above examples, great strides have been made in optimizing the use of collagen as a building block for tissue engineering. While in vitro polymerization is limited by the harsh processes involved in collagen extraction, several ways to modify and crosslink collagen to make up for the weak in vitro mechanical properties have been suggested. The bioactivity, adhesion sites, and signaling imparted by collagen is unique and undeniably almost a prerequisite for regenerating any tissue. However, on the flip side, most crosslinking methods represent a compromise between the degree of disruption of collagen structure and maximum tunability of the polymer.

While it is impossible to draw conclusions based on the concentrations and pH ranges used to synthesize collagen-based hydrogels, Table 1 summarizes the concentrations of collagen used based on the tissue of interest. It is fair to conclude that every modification approach comes with its drawbacks, and suggests that no one method is going to be the cure-all that resets the balance between loss of bioactivity and improvement of mechanical properties; unless a way to preserve the native collagen structure during purification and scaffold fabrication is established.

The addition of GAGs to collagen hydrogels has expanded the possibilities for the application of collagen hydrogels from tissue engineering to drug delivery. GAGs not only act as a tunable component of the gels, but also represent a more physiologically relevant ECM biomimic that is capable of modulating collagen fibrillogenesis, hydrogel properties and swelling, as well as guide cell behavior. Strides have been made in incorporating GAGs and collagen in spatially controlled hydrogels and innovative crosslinking strategies have allowed the preservation of fibrillar collagen in hydrogels that are still tunable without modification of the collagen. Based on the type of GAG used, different functionalities can be exploited to engineer biomimetics that closely resemble the tissue microenvironment in vivo, as well as allow for sustained release of growth factors and provide molecular cues that guide regeneration. Combinatorial approaches to biomaterials can thus provide higher control of cell function as well as open an array of new treatments for tackling complicated diseases. This review attempts to show examples of how far blended hydrogels of collagen and have come, but does not touch upon other blends of collagen hydrogels such as those blended with synthetic polymers or other natural polymers such as fibrin due to space constraints. One thing that does stand out, however, is that researchers may want to consider using a combination of not only biomaterials such as collagen and GAGs, but also cells and growth factors in order to modulate biophysical cell–gel interactions and develop successful efficacious treatments for regenerative medicine and drug delivery.

## Figures and Tables

**Figure 1 bioengineering-07-00156-f001:**
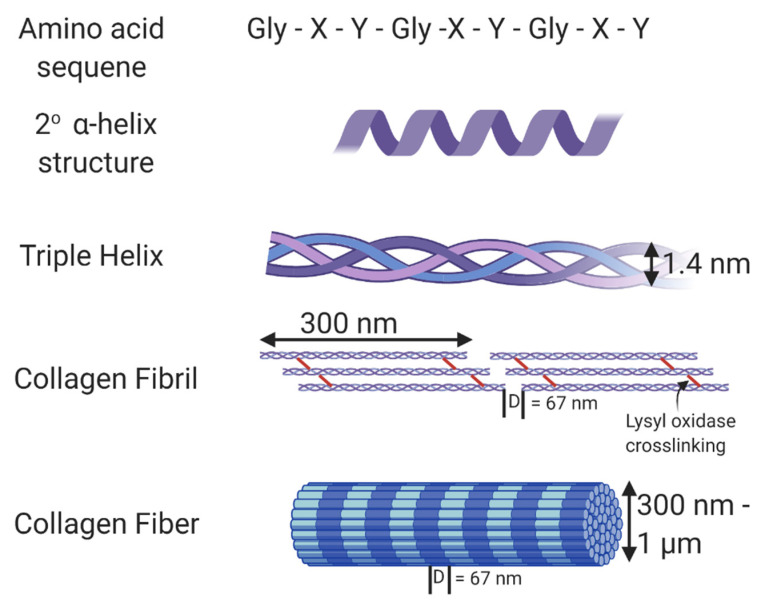
Structure of collagen. The amino acid sequence of collagen consists of Gly-Xaa-Yaa repeats, with Xaa and Yaa commonly occupied by proline and hydroxyproline. This unique sequence allows collagen to form an α helix secondary structure. Fibrillar collagen is a triple helix containing crosslinks formed through the action of lysyl oxidase. In vivo, these collagen fibrils form fibers with varying thickness and a D-banding pattern of 67 nm. Made using Biorender.

**Figure 2 bioengineering-07-00156-f002:**
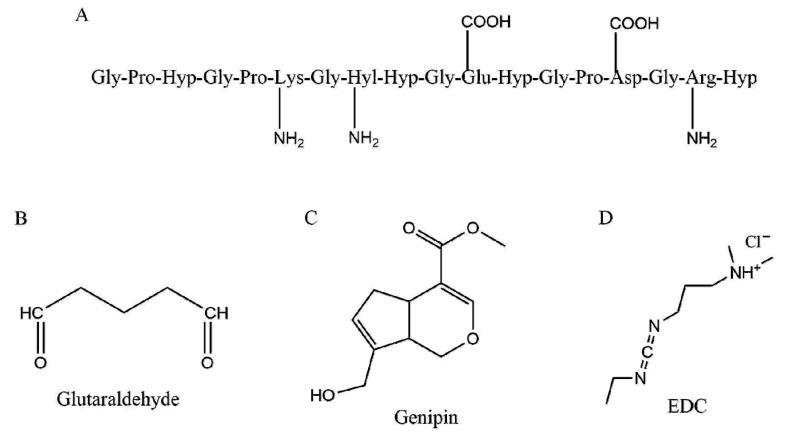
Chemical crosslinking of collagen. Crosslinks are primarily introduced in collagen by targeting carboxylic acid groups on Asp and Glu residues, or amines on Lys, Hyl, and Arg groups (**A**–**D**) show some common crosslinkers used to modify collagen.

**Figure 3 bioengineering-07-00156-f003:**
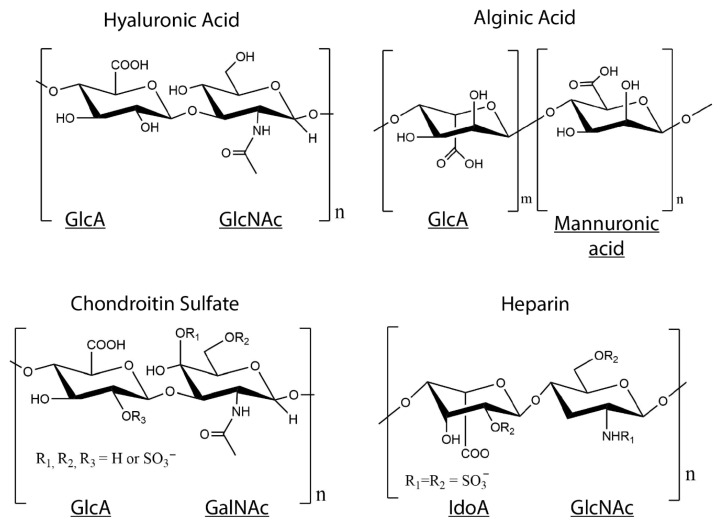
Structures of glycosaminoglycans (GAGs). Disaccharide monomer repeats of glucuronic acid (GlcA) or its epimer iduronic acid (IdoA), and N-acetylgalactosamine (GalNAc) or N-acetylglucosamine (GlcNAc) form the backbone of GAGs, except alginate, which is a GAG analog with GlcA and mannuronic acid repeats. Chondroitin sulfate and heparin are post-translationally sulfated and contain SO_3_^−^ groups on some disaccharide units.

**Table 1 bioengineering-07-00156-t001:** Examples of collagen-based hydrogels and their applications.

**Collagen Hydrogels with No Crosslinkers**			
**Hydrogel**	**Collagen Concentration and Temperature**	**Application**	**Reference**
Collagen type I	3 mg/mL, 37 °C 8 mg/mL, 37 °C 7 mg/mL, 37 °C 3.45 mg/mL, 37 °C	3D test bed for drug testing	[41]
		3D tumor model	[43,51]
		Stem cell differentiation	[47,48,49,50]
Electrochemically or magnetically aligned collagen	7 mg/mL, 37 °C 4 mg/mL, 37 °C 5 mg/mL, 37 °C 4 mg/mL, 30 °C 3 mg/mL, 25 °C	Tendon tissue engineering	[54,55]
		Cartilage tissue engineering	[56]
		Corneal tissue engineering	[57,58]
		Neural tissue engineering	[61]
Collagen type I and/or type II	4 mg/mL, 37 °C	Cartilage tissue engineering	[44,45,46,68]
Concentrated/compressed collagen	2 mg/mL, 10 mg/mL, 15 mg/mL, 25 °C	Dermal tissue engineering	[63,64]
**Crosslinked Collagen Hydrogels**			
**Hydrogel**	**Collagen Concentration and Temperature**	**Application**	**Reference**
EDC crosslinked collagen	6.33 mg/mL, 4 °C	Corneal tissue engineering	[67]
Genipin crosslinked collagen	2 mg/mL, 37 °C 6 mg/mL	Cartilage tissue engineering, stem cell differentiation	[69,72]
Dehydrothermal or UV crosslinked collagen	Unknown; Unknown	Vascular tissue engineering	[86]
		Tendon tissue engineering	[88]
Thiol crosslinked collagen	1% wt/v, 37 °C 3 mg/mL, 37 °C	Cardiovascular tissue engineering	[79]
		Liver regeneration	[113]
		Skin tissue engineering	[80]
**Collagen HA Hydrogels**			
**Hydrogel**	**Collagen Concentration and Temperature**	**Application**	**Reference**
Sulfated HA–collagen	1 mg/mL, 37 °C 0.5 mg/mL, 37 °C 1 mg/mL, 37 °C	Vascular tissue engineering	[105]
		Skin tissue engineering	[106]
		Bone tissue engineering	[107,108,109,110]
Thiolated HA–collagen IPN	4 mg/mL, 37 °C	Vocal fold tissue engineering	[114]
HA hydrazine, HA aldehyde–collagen IPN	2.5 mg/mL, 37 °C	Mimic in vivo microenvironment	[126]
Photocrosslinked HA–collagen IPN	3 mg/mL, 37 °C 3 mg/mL, 37 °C	Regenerative medicine	[115,116]
		Neural tissue engineering	[117]
EDC crosslinked HA–collagen	0.5 wt%, 1 wt% 5 mg/mL, 25 °C 1 mg/mL, 37 °C 6 mg/mL, 37 °C	Stem cell differentiation	[118,119]
		Cartilage tissue engineering	[60,120]
		Dermal tissue engineering	[123,125]
		Corneal tissue engineering	[124]
HA aldehyde–aminooxy PEG-collagen	50, 100, 200 μg/mL	Neural tissue engineering	[127]
AAD modified HA-collagen	8 mg/mL, 25 °C	Cartilage tissue engineering	[128]
**Collagen CS Hydrogels**			
**Hydrogel**		**Application**	**Reference**
Photocrosslinked CS–collagen IPN	5 mg/mL, 37 °C	Cartilage tissue engineering	[129]
Dehydrothermal crosslinked CS–collagen	2.6 mg/mL, 25 °C	Cartilage and dermal tissue engineering	[130]
Genipin crosslinked CS–collagen	1 mg/mL, 37 °C	Cartilage tissue engineering	[131]
EDC crosslinked CS–collagen	2.5 mg/mL 11, 8, 6, 4, 3 mg/mL	Dermal tissue engineering	[132,133]
		Cartilage tissue engineering	[60]
		Neural tissue engineering	[134,135]
Non crosslinked CS–collagen	4 mg/mL, 37 °C	Cartilage tissue engineering	[137]
**Collagen Heparin Hydrogels**			
**Hydrogel**	**Collagen Concentration and Temperature**	**Application**	**Reference**
Non crosslinked Heparin-collagen	4 mg/mL, 25 °C	Vascular tissue engineering	[139]
EDC crosslinked Heparin–collagen	2.5 mg/mL, 37 °C	Bone tissue engineering	[143]
starPEG–heparin–collagen	unknown	Cell instruction and differentiation	[142]
**Collagen–Alginate hydrogels**			
**Hydrogel**	**Collagen Concentration and Temperature**	**Application**	**Reference**
CaCl_2_ crosslinked alginate–collagen IPN	3 mg/mL, 37 °C 2.5 mg/mL, 37 °C 5 mg/mL, 37 °C 1 mg/mL, 37 °C 2 mg/mL, 37 °C	3D tumor model	[146]
		Neural tissue engineering	[147]
		Vocal fold tissue engineering	[148]
		Cartilage tissue engineering	[149,150,151]
		Corneal tissue engineering	[153]

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
