# Peer review of "Best of Both Hydrogel Worlds: Harnessing Bioactivity and Tunability by Incorporating Glycosaminoglycans in Collagen Hydrogels"

_bioengineering, 2020, doi:10.3390/bioengineering7040156_

Round 1

Reviewer 1 Report

The present manuscript explores the effect of glycosaminoglycans in collagen hydrogels. The objective seems interesting with potential application in biomedical engineering. However, revision is required and some aspects should be taken into account:

  • I miss a better justification for the behaviour of collagen vs. gelatin. Would all the methods described also be valid in the case of gelatin?
  • How do isolation and manufacturing affect the denaturation of collagen and, consequently, the final properties and microstructure of the hydrogel?
  • There are a large number of methods of manufacturing hydrogels. A more in-depth study, perhaps including a table, of the manufacturing methods of collagen-based hydrogels with their properties and applications would be welcome. How would you make a gel stable at 37 ° C without a crosslinking agent?
  • More information about the concentrations (of each biopolymer including crosslinking agents), pH values and manufacturing temperatures of collagen-GAG hydrogels is lacking.

Author Response

The present manuscript explores the effect of glycosaminoglycans in collagen hydrogels. The objective seems interesting with potential application in biomedical engineering. However, revision is required and some aspects should be taken into account:

  1. I miss a better justification for the behaviour of collagen vs. gelatin. Would all the methods described also be valid in the case of gelatin?

We thank the reviewer for this comment. We have now clarified in the paper that (Line 112) “Since gelatin is composed of denatured collagen and does not form fibrils despite forming hydrogels, to narrow the scope of the review to fibril forming collagen, we have not included studies with gelatin. Interested readers are directed to a review by Gorgieva and Kokol that discusses multiple modes of crosslinking and the effects on biophysical properties of both collagen and gelatin”. Moreover, in the paragraph “When collagen is denatured, such as in the form of gelatin ..” that starts on line 82, we mention that gelatin interacts with different ligands compared to native fibrillar collagen.  

Since gelatin is denatured and preserving fibril formation therefore is not a consideration anymore, the wide array of modifications for gelatin that exist were beyond the scope of this review. We focused on hydrogels that retained at least partial fibril forming capacity.

  1. How do isolation and manufacturing affect the denaturation of collagen and, consequently, the final properties and microstructure of the hydrogel?

We thank the reviewer for this question. We have added the following text to the paper to address it: (Line 137) “The hydroxylation of lysine and proline residues combined with lysyl oxidase crosslinking makes the production of functional collagen using traditional recombinant technology challenging. Researchers have therefore relied on extracting collagen from other species. Two main methods for extraction of collagen exist - acid solubilization at low pH values using an organic acid such as acetic acid, and a combination of salt precipitation with enzymatic extraction (pepsin digestion). Acid-solubilized collagen largely maintains its telopeptide regions that are critical sites for crosslinking, and in fact this extraction co-isolates a small number of multimers with crosslinks intact. While pepsin digestion results in fully cleaved terminal non-helical regions that contain the intermolecular crosslinks. Pepsin-digested collagen results in a more soluble form of collagen resulting in higher yields, but also a form of collagen that polymerizes more slowly and exhibits decreased storage modulus values compared to acid-solubilized collagen [33,34] possibly since telopeptides play a strong role in fibril nucleation and are the locations of native crosslinks”

  1. There are a large number of methods of manufacturing hydrogels. A more in-depth study, perhaps including a table, of the manufacturing methods of collagen-based hydrogels with their properties and applications would be welcome. How would you make a gel stable at 37 ° C without a crosslinking agent?

We thank the reviewer for pointing out this important consideration. We have now added methods to make a stable collagen gel at 37 °C without the addition of crosslinkers. This method is universally used for manufacturing of collagen hydrogels. However, parameters such as concentration, pH, and temperature affect fibrillogenesis and downstream mechanical properties of the hydrogels. To bring this to the reader’s attention, we have added the following two paragraphs to the manuscript (Line 130):

“Collagen hydrogels are synthesized by neutralizing acid-solubilized collagen using concentrated buffers [10X phosphate buffered saline (PBS), 10X cell culture medium, 10X Hank’s balanced salt solution (HBSS), etc] to bring the ionic strength of the solution to 1X followed by the addition of neutralization agents [NaOH, HCl, and HEPES] and other reagents (water, 1X medium, 1X PBS) to initiate fibril self-assembly at a near physiological pH and polymerization temperatures of 37 °C. While this recipe for hydrogel preparation is universal, a closer look at the literature reveals that collagen hydrogel properties are highly dependent on collagen source (rat tail tendon, bovine skin, porcine, etc) [32] and method of extraction [33]. The hydroxylation of lysine and proline residues combined with lysyl oxidase crosslinking makes the production of functional collagen using traditional recombinant technology challenging. Researchers have therefore relied on extracting collagen from other species. Two main methods for extraction of collagen exist - acid solubilization at low pH values using an organic acid such as acetic acid, and a combination of salt precipitation with enzymatic extraction (pepsin digestion). Acid-solubilized collagen largely maintains its telopeptide regions that are critical sites for crosslinking, and in fact this extraction co-isolates a small number of multimers with crosslinks intact. While pepsin digestion results in fully cleaved terminal non-helical regions that contain the intermolecular crosslinks. Pepsin-digested collagen results in a more soluble form of collagen resulting in higher yields, but also a form of collagen that polymerizes more slowly and exhibits decreased storage modulus values compared to acid-solubilized collagen [33,34] possibly since telopeptides play a strong role in fibril nucleation and are the locations of native crosslinks

In a recent review investigating initial collagen solution concentration, pH, temperature, and ionic strength, it was noted that the rate of polymerization, compressive modulus, fibril diameter and pore size in gels formed from acid solubilized collagen all influenced hydrogel properties [31]. Collagen concentration influences mechanical properties of the hydrogels, thereby influencing cell behavior [35,36]. This is perhaps not surprising when reflecting upon the fact that ECM compositions vary as does degree of crosslinking across tissue types. Predictably, reaction kinetics for collagen fibril assembly are temperature dependent, therefore, collagen molecules self-assemble more rapidly at higher temperatures. However, fibrils polymerized at higher temperatures show a lower number of bundled fibrils that are less ordered, consequently affecting the mechanical and structural properties of the hydrogel [37,38]. Fibril assembly initiates as soon as the collagen solution is neutralized regardless of temperature, hence, most groups work with collagen solutions on ice to slow the rate of polymerization until the hydrogel components are well mixed. pH of the solution also greatly affects structural and mechanical properties of the fibrils by modulating electrostatic interactions [39], thus adding complexity of working with collagen hydrogels. A strong positive correlation between pH and compressive strength exists; however, for physiological encapsulation, pH of hydrogels is limited to between 7.4 – 8.4 to maintain cell viability [40]. Both, temperature and pH may influence the ratio of monomeric collagen to crosslinked multimeric collagen isolated from a different tissue type, or the same tissue type across species or from different aged animals or animals with a different environmental upbringing. Also, ECM composition variation may result in other types of collagen and accessory molecules co-purifying with the collagen type 1. All of these factors can influence the ultimate collagen hydrogel properties. Reviewers are directed towards the in-depth review by Antoine et al. for further reading on fabrication parameters [31]. A few examples for applications of collagen hydrogels synthesized through neutralization-based self-assembly follow.”

  1. More information about the concentrations (of each biopolymer including crosslinking agents), pH values and manufacturing temperatures of collagen-GAG hydrogels is lacking.

We thank the reviewer for bringing this to our attention. Table 1 has been modified to include the concentration, pH, and temperatures of the collagen-GAG hydrogels.

Reviewer 2 Report

In this review, the authors summarized the recent advances in the development of collagen-based hydrogels and collagen-glycosaminoglycan blend hydrogels for biomedical research. Meanwhile, the shortcomings of using collagen in isolation and the advantages of incorporating GAGs in the hydrogels were also discussed. The review provided the versatility and highly translational value of using collagen blended with GAGs as hydrogels for biomedical engineering applications. If the following problems are properly addressed, this paper could be published.

  1. It was suggested to revise some Chinese symbols appearing in the manuscript.
  2. Please systematically explain how the collagen solution concentration, pH, and temperature influence the collagen hydrogel properties.
  3. The authors list several crosslinkers for crosslinking collagen, such as EDC/N‐hydroxysuccinimide (EDC/NHS), glutaraldehyde, and genipin. How about the research progress using biodegradable crosslinkers such as linkers with disulfide?
  4. Please explain how the GAGs, crosslinker and the degree of crosslinking influence the biological activity of collagen-GAG hydrogels.

Author Response

Reviewer 2:

In this review, the authors summarized the recent advances in the development of collagen-based hydrogels and collagen-glycosaminoglycan blend hydrogels for biomedical research. Meanwhile, the shortcomings of using collagen in isolation and the advantages of incorporating GAGs in the hydrogels were also discussed. The review provided the versatility and highly translational value of using collagen blended with GAGs as hydrogels for biomedical engineering applications. If the following problems are properly addressed, this paper could be published.

  1. It was suggested to revise some Chinese symbols appearing in the manuscript.

We are sorry for the confusion, but we carefully went through the paper, and could not find any Chinese symbols in the manuscript.  Perhaps this is due to a Microsoft Word compatibility issue.

  1. Please systematically explain how the collagen solution concentration, pH, and temperature influence the collagen hydrogel properties.

We thank the reviewer for asking this important question on parameters affecting collagen hydrogel properties. We have added 2 paragraphs (Line 130) on collagen fabrication parameters and how they affect fibril formation.

“Collagen hydrogels are synthesized by neutralizing acid-solubilized collagen using concentrated buffers [10X phosphate buffered saline (PBS), 10X cell culture medium, 10X Hank’s balanced salt solution (HBSS), etc] to bring the ionic strength of the solution to 1X followed by the addition of neutralization agents [NaOH, HCl, and HEPES] and other reagents (water, 1X medium, 1X PBS) to initiate fibril self-assembly at a near physiological pH and polymerization temperatures of 37 °C. While this recipe for hydrogel preparation is universal, a closer look at the literature reveals that collagen hydrogel properties are highly dependent on collagen source (rat tail tendon, bovine skin, porcine, etc) [32] and method of extraction [33]. The hydroxylation of lysine and proline residues combined with lysyl oxidase crosslinking makes the production of functional collagen using traditional recombinant technology challenging. Researchers have therefore relied on extracting collagen from other species. Two main methods for extraction of collagen exist - acid solubilization at low pH values using an organic acid such as acetic acid, and a combination of salt precipitation with enzymatic extraction (pepsin digestion). Acid-solubilized collagen largely maintains its telopeptide regions that are critical sites for crosslinking, and in fact this extraction co-isolates a small number of multimers with crosslinks intact. While pepsin digestion results in fully cleaved terminal non-helical regions that contain the intermolecular crosslinks. Pepsin-digested collagen results in a more soluble form of collagen resulting in higher yields, but also a form of collagen that polymerizes more slowly and exhibits decreased storage modulus values compared to acid-solubilized collagen [33,34] possibly since telopeptides play a strong role in fibril nucleation and are the locations of native crosslinks

In a recent review investigating initial collagen solution concentration, pH, temperature, and ionic strength, it was noted that the rate of polymerization, compressive modulus, fibril diameter and pore size in gels formed from acid solubilized collagen all influenced hydrogel properties [31]. Collagen concentration influences mechanical properties of the hydrogels, thereby influencing cell behavior [35,36]. This is perhaps not surprising when reflecting upon the fact that ECM compositions vary as does degree of crosslinking across tissue types. Predictably, reaction kinetics for collagen fibril assembly are temperature dependent, therefore, collagen molecules self-assemble more rapidly at higher temperatures. However, fibrils polymerized at higher temperatures show a lower number of bundled fibrils that are less ordered, consequently affecting the mechanical and structural properties of the hydrogel [37,38]. Fibril assembly initiates as soon as the collagen solution is neutralized regardless of temperature, hence, most groups work with collagen solutions on ice to slow the rate of polymerization until the hydrogel components are well mixed. pH of the solution also greatly affects structural and mechanical properties of the fibrils by modulating electrostatic interactions [39], thus adding complexity of working with collagen hydrogels. A strong positive correlation between pH and compressive strength exists; however, for physiological encapsulation, pH of hydrogels is limited to between 7.4 – 8.4 to maintain cell viability [40]. Both, temperature and pH may influence the ratio of monomeric collagen to crosslinked multimeric collagen isolated from a different tissue type, or the same tissue type across species or from different aged animals or animals with a different environmental upbringing. Also, ECM composition variation may result in other types of collagen and accessory molecules co-purifying with the collagen type 1. All of these factors can influence the ultimate collagen hydrogel properties. Reviewers are directed towards the in-depth review by Antoine et al. for further reading on fabrication parameters [31]. A few examples for applications of collagen hydrogels synthesized through neutralization-based self-assembly follow.”

  1. The authors list several crosslinkers for crosslinking collagen, such as EDC/N‐hydroxysuccinimide (EDC/NHS), glutaraldehyde, and genipin. How about the research progress using biodegradable crosslinkers such as linkers with disulfide?

Thank you for pointing us in this direction, we apologize for overlooking this important modification. We have now added the following paragraph on disulfide and thiol crosslinked collagen (Line 327):

“Thiolation of gelatin is a popular method to add thiols to gelatin for future crosslinking with other thiols to form disulfides or with acrylates through Michael type addition [78]. Along those lines, researchers have attempted to thiolate collagen to use the functional thiol for controlling the degree of crosslinking and increasing the stability of collagen hydrogels by adding other biopolymers like polyethylene glycol (PEG) or bioactive compounds to it. There are multiple ways to introduce thiols and disulfide linkages in collagen. One method to form thiolated collagen is by reacting collagen with succinic anhydride to yield carboxylated collagen (Col-COOH), followed by amidation with 2-mercaptoethylamine hydrochloride (MEA) [79] or to simply react with N-succinimidyl S-acetylthioacetate [80]. Another method is reacting it in an imidazole aqueous solution or dimethyl sulfoxide and reacting with γ-thiobutyrolactone [81,82]. Thiolating collagen provides the opportunity to tune collagen hydrogels to the required tissue while improving its stability. However, it is unclear the degree to which the collagen retains its fibril forming abilities upon modifications with thiol containing agents, as microstructural characterization was not performed in these studies. The Tanabe research group showed that disulfide crosslinked collagen could preserve partial helix structure [83], suggesting that this modification reduces fibril forming ability of collagen. Future studies characterizing the effect of loss of fibril formation on biocompatibility will be needed to provide insights on the degree of modification ideal for use in hydrogel systems”

  1. Please explain how the GAGs, crosslinker and the degree of crosslinking influence the biological activity of collagen-GAG hydrogels.

In order to address this more clearly in the manuscript, we added the following paragraph to it (line 410): “As with crosslinked collagen hydrogels, parameters such as concentration, crosslinking density, and retention of bioactivity are important fabrication parameters for the design of collagen-GAG hydrogels. High concentrations of GAGs inhibit fibril formation [101], limiting the amount of GAGs that can be added to the hydrogel without negatively affecting fibril formation. Degree of modification and crosslinking density directly correlate with increased stiffness and stability of the hydrogels, which can be used to design hydrogels matching tissue microenvironments. However, care needs to be taken to not over-substitute GAGs, since higher degrees of modifications result in the loss of bioactivity [102]. Therefore, a balance between concentration, modification degree, and bioactivity needs to be found for the design of GAG-collagen hydrogels. Designing successful hydrogel candidates consequently requires careful understanding of the required tissue outcome and parameters important for regeneration. The next section includes examples of collagen-GAG hydrogels organized by the type of GAG used.”
